# Angiographic Predictors for Repeated Revascularization in Patients with Intermediate Coronary Lesions

**DOI:** 10.3390/biomedicines12122825

**Published:** 2024-12-12

**Authors:** Yong-Kyun Kim, Soon-Ho Kwon, Young-Hoon Seo, Ki-Hong Kim, Taek-Geun Kwon, Jang-Ho Bae

**Affiliations:** Division of Cardiology, Department of Internal Medicine, Konyang University Hospital, Daejeon 35365, Republic of Korea; 200653@kyuh.ac.kr (Y.-K.K.); beartank@kyuh.ac.kr (S.-H.K.); syhheart@kyuh.ac.kr (Y.-H.S.); iquitoskihong@kyuh.ac.kr (K.-H.K.); tgkwon@kyuh.ac.kr (T.-G.K.)

**Keywords:** intermediate coronary lesion, coronary artery stenosis, prognosis

## Abstract

**Background:** Intermediate lesions (ILs) present challenges in making therapeutic decisions. This study aimed to determine the practical coronary angiographic predictors for revascularization in patients with ILs who underwent repeated angiograms. **Methods:** This study was a retrospective single-center study. The study subjects were divided into two groups according to their target lesion revascularization (TLR) during the follow-up period: the TLR (+) group (n = 135, 30.9%) and the TLR (−) group (n = 302, 69.1%). We evaluated the angiographic characteristics of ILs such as the presence of branches, luminal irregularity, tortuosity, ulcer/erosion, haziness, and calcification in the ILs, with an average follow-up of 34.2 ± 32.0 months. **Results:** The TLR (+) group had higher percentage of diameter stenoses (47.3 ± 13.5% vs. 44.2 ± 12.2%, *p* = 0.006) than the TLR (−) group, whereas the lesion length of the ILs showed no significant differences between the two groups. The prevalence of branches (79.0% vs. 69.1%, *p* = 0.018) and haziness (4.3% vs. 2.6%, *p* < 0.001) was higher in the ILs of the TLR (+) group than those of the TLR (−) group. Therefore, the angiographic predictors for the TLR of ILs were haziness (hazard ratio = 2.126, 95% confidence interval = 1.240–3.644, *p* = 0.006) and % diameter stenosis (DS) ≥ 60% (hazard ratio = 1.025, 95% confidence interval = 1.013–1.037, *p* < 0.001). **Conclusions:** Angiographic haziness and % DS > 60% were the independent angiographic predictors for TLR in patients with ILs. Our study is the first to present the angiographic findings of vulnerable plaques of ILs. Further studies such as intravascular imaging or physiologic studies should be strongly considered before making treatment decisions in ILs when such angiographic features are observed.

## 1. Introduction

Intermediate lesions (ILs) are frequently seen during coronary angiograms [1,2,3]. Fortunately, ILs show excellent clinical outcomes [4,5]. However, we still have difficulty in making therapeutic decisions in patients with ILs. Although fractional flow reserve (FFR) is regarded as the gold standard upon which to make therapeutic decisions in ILs during coronary angiograms [3,6,7,8,9,10,11,12,13], the FFR penetration rate is quite low in real-world practices [14,15]. Furthermore, a negative FFR, that is, an FFR value of 0.80 or more, constitutes about two-thirds of non-culprit ILs, and most ILs show favorable clinical outcomes [7]. Intravascular ultrasound (IVUS) studies have shown that there are no definite minimal luminal area (MLA) criteria upon which to make therapeutic decisions because not all patients with ILs have the same vessel diameter and lesion location. Therefore, the IVUS penetration rate is also quite low [15,16], although its importance in percutaneous coronary intervention (PCI) is continuously reported in terms of mortality and other adverse events [8]. However, some IVUS features are known as predictors for adverse events, including revascularization, such as thin-cap fibroatheroma, MLA, and plaque burden [17,18,19,20].

Both FFR and IVUS play important roles in the treatment decision of ILs [8,10]. However, their low penetration rates in real-world use due to being costly, relatively unfamiliar, and time consuming have limited their clinical implication in understanding and treating ILs in daily practice [14,15,16].

Most interventional cardiologists have experience with ILs progressing into significant lesions during coronary angiograms, which then need revascularization, especially in patients with ILs who underwent repeated coronary angiograms due to recurring angina. Therefore, we tried to identify the practical coronary angiographic predictors for revascularization in patients with ILs who underwent repeated angiograms without FFR or IVUS studies.

## 2. Study Population and Methods

### 2.1. Study Population

This study was a retrospective single-center study. We reviewed the medical records of a total of 667 patients who underwent at least two coronary angiograms (CAGs) due to ischemic heart disease or other cardiovascular diseases between January 2008 and December 2020 in Konyang University Hospital, Daejeon, Republic of Korea. A total of 667 patients underwent 1517 CAGs during the study period. We excluded 230 patients who did not have any ILs and underwent a staged procedure. Thus, a total of 437 patients with 627 ILs were finally enrolled in this study.

We reviewed the patients’ demographic, laboratory, and angiographic findings. An IL was defined as angiographic stenosis from 30% to 70% based on visual estimation [21,22].

The study subjects were divided into two groups according to their target lesion revascularization (TLR) during the follow-up period: the TLR (+) group and the TLR (−) group. TLR was defined as PCI of the IL due to any ischemic event such as recurrent angina, ischemic evidence based on cardiac enzymes, regional wall motion abnormalities due to IL progression, etc.

This study was approved by the Institutional Review Board of Konyang University Hospital (2023-11-008) and performed in accordance with the ethical guidelines of the 1975 Declaration of Helsinki. No informed consent was required.

### 2.2. Angiographic Analysis

We assessed the angiographic characteristics of ILs such as the presence of branches, luminal irregularity, tortuosity, ulcer/erosion, haziness, or calcification in the ILs on a coronary angiogram. A branch was considered present if a branch vessel was larger than 1.0 mm originating from the IL. IL irregularity was considered present when the lumen surface of the IL was jagged or not smooth. IL tortuosity was considered present when the IL could be bent by more than 45° angle. Ulcer/erosion was considered present when a small crater did not extend beyond the lumen of the IL. Haziness was considered present if a reduction in contrast density or an indistinct vessel border was observed on the coronary angiogram. Calcification was considered present if readily apparent densities were seen in the IL.

Two researchers (YKK and SHK), who had more than 10 years of experience working with coronary angiograms, reviewed and determined the angiographic stenosis with lesion characteristics. A senior cardiologist (JHB), who had more than 27 years of experience working with coronary angiograms, gave a second opinion if the two researchers had differing opinions about the degree of IL stenosis or the IL characteristics.

### 2.3. Statistical Analysis

Continuous variables are presented as mean ± standard deviation, and categorical variables are presented as numbers and percentages. In the univariable analysis, Chi-square test or Fisher’s exact test was performed to examine the differences between groups in the categorical variables, and unpaired Student’s *t*-test was used to assess differences between groups in the continuous variables. Logistic regression analyses were performed to determine the clinical and angiographic correlates of outcome. A multivariable Cox’s proportional hazard analysis was performed to investigate the independent predictors of TLR. Clinical patient demographics and angiographic characteristic variables that showed significant relationships with TLR in the univariable analysis with *p*-value < 0.10 were entered into the multivariable analysis. The Kaplan–Meier method with the log rank test was used to statistically assess the TLR-free rate according to each angiographic feature.

The statistical data were processed using the SPSS program (version 18.0, Chicago, IL, USA), and a *p*-value < 0.05 was considered statistically significant.

## 3. Results

There were 627 ILs in the 437 patients who underwent repeated coronary angiograms. The time interval between the index and follow-up CAGs was 34.2 ± 32.0 months in this study. Out of all the study patients, 135 patients (30.9%) with ILs underwent TLR, and out of the 627 ILs, 157 ILs (25.0%) underwent TLR.

### 3.1. Patient Demographics (Table 1)

The TLR (+) group had higher serum total cholesterol, LDL cholesterol, and hemoglobin levels but lower DM prevalence and WBC counts than the TLR (−) group. In the TLR (+) group, 100 patients (74.1%) out of a total of 135 underwent PCI due to severe stenotic culprit lesions that were not related to the IL during the index procedure, whereas in the TLR (−) group, 214 patients (70.9%) out of a total of 302 patients underwent PCI. The TLR (+) group showed more use of RAS inhibitors and beta blockers than the TLR (−) group during the index procedure.

The other demographics showed no significant differences between the two groups in terms of age, gender, other risk factors, kidney function, high sensitivity C-reactive protein, ejection fraction, multi-vessel disease, and other medications during the index procedure.

**Table 1 biomedicines-12-02825-t001:** Demographics of the patients with intermediate lesions.

Variables	TLR (−) n = 302 (69.1%)	TLR (+) n = 135 (30.9%)	Total n = 437 (100%)	*p*-Value
FU duration, month	33.0 ± 32.0	36.8 ± 32.0	34.2 ± 32.0	0.254
Age, years	63.4 ± 11.0	62.6 ± 10.5	63.2 ± 10.8	0.443
Men, n (%)	213 (70.5)	96 (71.1)	309 (70.7)	0.902
Hypertension, n (%)	207 (68.5)	87 (64.4)	294 (67.3)	0.399
Diabetes mellitus, n (%)	128 (42.4)	40 (29.6)	168 (38.4)	0.011
Smoking, n (%)	74 (30.8)	34 (30.6)	108 (30.8)	0.969
Diagnosis, n (%)				
Stable angina	165 (54.6)	66 (48.9)	231 (52.9)	0.266
ACS	122 (40.4)	65 (48.1)	187 (42.8)	0.130
Lipid profile				
Total cholesterol, mg/dL	166.7 ± 47.0	181.2 ± 47.9	171.3 ± 47.7	0.004
Triglyceride, mg/dL	163.1 ± 131.9	169.6 ± 99.5	165.1 ± 122.6	0.614
HDL cholesterol, mg/dL	45.9 ± 35.1	44.3 ± 24.4	45.4 ± 32.1	0.647
LDL cholesterol, mg/dL	76.9 ± 28.3	80.7 ± 28.7	114.7 ± 37.6	0.004
Creatinine, mg/dL	1.31 ± 1.62	1.21 ± 1.53	1.28 ± 1.59	0.545
eGFR, mL/min/1.73 m^2^	77.1 ± 26.6	79.1 ± 22.9	77.7 ± 25.5	0.452
Fasting glucose, mg/dL	134.8 ± 65.7	127.1 ± 57.3	132.4 ± 63.2	0.244
WBC, /μL	8329.4 ± 3428.9	7655.4 ± 2426.7	8122.3 ± 3167.5	0.020
Hb, g/dL	13.2 ± 2.1	13.8 ± 3.6	13.4 ± 2.7	0.037
HbA1C, %	7.3 ± 4.9	6.7 ± 1.3	7.1 ± 4.2	0.301
Hs C-reactive protein, mg/L	0.86 ± 6.20	0.42 ± 1.05	0.73 ± 5.24	0.482
Ejection fraction, %	61.6 ± 11.4	63.1 ± 9.8	62.1 ± 10.9	0.192
Multi-vessel disease, n (%)	173 (57.3)	87 (64.4)	260 (59.5)	0.159
LM disease, n (%)	14 (4.6)	6 (4.4)	20 (4.6)	0.930
Previous PCI, n (%)	33 (10.9)	16 (11.9)	49 (11.2)	0.777
Previous CABG, n (%)	2 (0.7)	1 (0.7)	3 (0.7)	1.000
Initial PCI site	287	123	410	0.896
LM, n (%)	9 (3.1)	1 (0.8)	10 (2.4)	
LAD, n (%)	130 (45.3)	55 (44.7)	185 (45.1)	
LCX, n (%)	66 (23.0)	30 (24.4)	96 (23.4)	
RCA, n (%)	82 (28.6)	37 (30.1)	119 (29.0)	
Medication				
Aspirin	300 (99.3)	135 (100.0)	435 (99.5)	1.000
P2Y12 inhibitor				0.402
Clopidogrel	226 (74.8)	104 (77.0)	330 (75.5)	
Ticagrelor	29 (9.6)	16 (11.9)	45 (10.3)	
Anticoagulation				0.427
Warfarin	1 (0.3)	0 (0.0)	1 (0.2)	
DOAC	5 (1.7)	0 (0.0)	5 (1.1)	
RASI				0.032
ACEi	18 (6.0)	16 (11.9)	34 (7.8)	
ARB	180 (59.6)	85 (63.0)	265 (60.6)	
BB	184 (60.9)	100 (74.1)	284 (65.0)	0.008
Statin	284 (94.0)	123 (91.1)	407 (93.1)	0.263

TLR, target lesion revascularization; FU, follow-up; ACS, acute coronary syndrome; HDL, high-density lipoprotein; LDL, low-density lipoprotein; eGFR, estimated glomerular filtration rate; WBC, white blood cell; Hb, hemoglobin; HbA1C, hemoglobin A1C; Hs C-reactive protein, high-sensitivity C-reactive protein; LM, left main; PCI, percutaneous coronary intervention; CABG, coronary artery bypass graft; LAD, left anterior descending artery; LCX, left circumflex artery; RCA, right coronary artery; P2Y12; DOAC, direct oral anticoagulant; RASI, renin–angiotensin system inhibitor; ACEi, angiotensin-converting enzyme inhibitor; ARB, angiotensin receptor blocker; BB, beta blocker.

### 3.2. Coronary Angiographic Findings (Table 2)

IL location during the index procedure showed no significant difference between the two groups. The TLR (+) group had a higher percentage of diameter stenoses (47.3 ± 13.5% vs. 44.2 ± 12.2%, *p* = 0.006) than the TLR (−) group, whereas lesion length of the ILs showed no significant differences between the two groups.

The prevalence of branches (79.0% vs. 69.1%, *p* = 0.018) and haziness (4.3% vs. 2.6%, *p* < 0.001) was higher in the ILs of the TLR (+) group than in those of the TLR (−) group, but no significant differences were found in the irregularity, tortuosity, ulcer/erosion, and calcification of the ILs between the two groups. The representative ILs with haziness were shown in Figure 1 which became severe stenotic and eventually need TLR. 

The ILs of the TLR (+) group became severely stenotic (85.8 ± 14.6% vs. 45.0 ± 15.7%, *p* < 0.001) compared with those of the TLR (−) group, which had a higher prevalence of acute coronary syndrome (47.7% vs. 36.8%, *p* = 0.016) during their follow-up CAGs. The ILs of the TLR (+) group were mostly (98.1%) treated with stents.

**Table 2 biomedicines-12-02825-t002:** Angiographic characteristics of the intermediate lesion.

Variables	TLR (−) 470 (75.0%)	TLR (+) 157 (25.0%)	Total 627 (100%)	*p*-Value
IL location				0.183
LAD, n (%)	164 (34.9)	63 (40.1)	227 (36.2)	
LCX, n (%)	100 (21.3)	35 (22.3)	135 (21.5)	
RCA, n (%)	203 (43.2)	56 (35.7)	259 (41.3)	
LM, n (%)	3 (0.6)	3 (1.9)	6 (1.0)	
IL site				0.494
Ostium, n (%)	9 (1.9)	5 (3.2)	14 (2.2)	
Proximal, n (%)	144 (30.6)	55 (35.0)	199 (31.7)	
Middle, n (%)	247 (52.6)	78 (49.7)	325 (51.8)	
Distal, n (%)	70 (14.9)	19 (12.1)	89 (14.2)	
Percent diameter stenosis (%)	44.2 ± 12.2	47.3 ± 13.5	44.9 ± 12.6	0.006
% DS ≥ 60%, n (%)	89 (18.9%)	46 (29.3)	135 (21.5)	
Lesion length, mm	18.4 ± 11.1	19.3 ± 10.9	18.6 ± 11.1	0.367
IL angiographic characteristics				
Branch	325 (69.1)	124 (79.0)	449 (71.6)	0.018
Irregularity	63 (13.4)	28 (17.8)	91 (14.5)	0.172
Tortuosity	143 (30.4)	44 (28.0)	187 (29.8)	0.569
Ulcer/erosion rupture	23 (4.9)	12 (7.6)	35 (5.6)	0.194
Calcium	106 (22.6)	37 (23.6)	143 (22.8)	0.793
Haziness	12 (2.6)	15 (4.3)	27 (4.3)	<0.001
FU diagnosis				
SAP	278 (59.8)	81 (52.3)	359 (57.9)	0.100
ACS	171 (36.8)	74 (47.7)	245 (39.5)	0.016
FU percent diameter stenosis	45.0 ± 15.7	85.8 ± 14.6	55.2 ± 23.5	<0.001
FU treatment				<0.001
Medication only	470	0 (0.0)		
+POBA		2 (0.5)		
+Stent		154 (98.1)		
+CABG		1 (0.6)		

TLR, target lesion revascularization; IL, intermediate lesion; LAD, left anterior descending artery; LCX, left circumflex artery; RCA, right coronary artery; LM, left main; % DS, percent diameter stenosis; FU, follow-up; SAP, stable angina pectoris; ACS, acute coronary syndrome; POBA, percutaneous old balloon angioplasty; CABG, coronary artery bypass graft.

### 3.3. Angiographic Predictors of IL for TLR

Branches and haziness occurred in 449/627 (71.6%) and 27/627 (4.3%) of ILs, respectively, and % diameter stenosis was ≥60% in 135/627 (21.5%) of ILs. The TLR rates for ILs with haziness, % diameter stenosis ≥ 60%, and branches were 55.6% (15/27), 34.1% (46/135), and 27.6% (124/449), respectively (Figure 2), while the TLR rate of ILs with branches and % diameter stenosis ≥ 60% was 16.3% (39/102). Thus, the number of ILs with features such as haziness, branches, and % diameter stenosis ≥ 60% was too small to analyze for this study.

The Kaplan–Meier curve analysis also showed that the TLR rates were significantly lower if there was no haziness, no branches, or % diameter stenosis < 60% in the ILs (Figure 3).

The multivariate analysis showed that the independent angiographic predictors for the TLR of ILs were haziness (hazard ratio = 2.126, 95% confidence interval = 1.240–3.644, *p* = 0.006) and % diameter stenosis ≥ 60% (hazard ratio = 1.025, 95% confidence interval = 1.013–1.037, *p* < 0.001) (Table 3).

## 4. Discussion

The main finding of this study is that the independent angiographic predictors for the TLR of ILs were angiographic haziness and DS ≥ 60% in patients with ILs, although the presence of branches also showed significantly higher TLR rates of ILs in the KM curve analysis.

Vulnerable plaques can be defined as a rupture prone to plaques or rapidly progressing plaques, thus requiring revascularization during follow-up visits [23]. It is clinically important because it may cause acute coronary syndrome, which can be fatal [24]. Few angiographic studies have been conducted to unveil vulnerable plaques because of their relatively poor accuracy in understanding the vessel lumen such as plaque composition, luminal narrowing, presence of calcium, plaque rupture, etc., compared with IVUS in patients with IL [25,26]. IVUS studies disclosed the presence of several features showing VP such as large lipid cores, thin fibrous caps, high levels of macrophage infiltration into the fibrous cap, high plaque burden (>70%), and small MLAs [20,27,28]. However, there are some limitations in its clinical implication because of the very low rate of vulnerable plaques, the high cost of IVUS, and the still quite low penetration rate of IVUS in real practice [14,16].

Currently, FFR is a very important tool in making decisions about ILs, especially regarding the need for revascularization [3]. However, most ILs have FFR values greater than 0.8, considered as FFR negative ILs, which reduces patient safety [7] due to most ACSs resulting from plaque rupture in non-stenotic coronary lesions. Thus, FFR itself has some limitations in predicting the future risk of IL, although it is still very helpful in making decisions about the current ischemia of ILs.

We previously reported excellent very long-term clinical outcomes in patients with ILs [4,5]. The TLR rate of ILs was quite similar to those of stented lesions in patients with ILs during a 10-year follow-up period [5]. It is therefore quite reasonable to retain the use of optimal medications to treat IL when we think about the costs, the risk of PCI, and the risk of bleeding from antiplatelet therapy. However, some ILs may still cause acute coronary syndrome via plaque rupture or rapid progression [2,29]. Therefore, the specific features of vulnerable plaques in ILs are still a big concern to cardiologists, and the identification of such features based on coronary angiograms without the use of IVUS would be beneficial in real practice. Our study is the first, as far as we know, to show the possible angiographic features suggestive of vulnerable plaques in patients with ILs.

Intraluminal haziness, which was defined in our study as a reduction in contrast density, or an indistinct vessel border on a coronary angiogram, or intraluminal filling defects in a vessel, or adjacent to a stenotic lesion, is a common finding during coronary angiography. There are various pathological causes that can cause intraluminal haziness or filling defects. Most of them can be caused by thrombi, but VP and spontaneous coronary artery dissection are also possible causes [30,31,32]. A recanalized coronary thrombus following thrombotic occlusion is a rare entity but also a possible cause [33]. In particular, haziness at sites of PCI is believed to represent dissection, luminal surface disruption, thrombus formation due to plaque rupture, or platelet deposition, which can be clearly differentiated by intravascular imaging such as intravascular ultrasound or optical coherence tomography [34,35]. Thus, previous studies on the etiology of intraluminal haziness can explain why haziness was an independent angiographic predictor for TLR of ILs.

The receiver operator characteristic curve analysis showed that the cutoff point of DS ≥ 60% had a sensitivity of 29.3% and a specificity of 81.1% for the prediction of TLR (area under the curve 0.565, *p* = 0.015). However, DS ≥ 60% is difficult to use as a prognostic factor of ILs for TLR because its sensitivity and HR are too low (HR = 1.025).

The Kaplan–Meier curves showed that the presence of branched ILs was a statistically significant factor for TLR. The coronary flow dynamic plays a central role in the progression of coronary artery disease. Abnormal endothelial wall shear stress (WSS) stimulates atherosclerosis and luminal remodeling, such as intimal thickening and increases in plaque burden [36,37,38]. VP formation and luminal thrombus can occur due to low WSS in areas with disturbed blood flow such as branched lesions. Thus, vascular geometry is an important factor in coronary artery disease. However, the diameter, size, and length of an IL branch can impact WSS, which is why a branched IL was not an independent factor for TLR in our study.

There are some limitations in this study: First, this study is a retrospective study. However, a prospective study with the same aims as our study would be very difficult to carry out, as the rate of IL occurrence is very low. Another limitation is that our study did not exam IVUS or FFR in these study subjects. However, the lack of other invasive examinations is a unique feature of our study. Yet another limitation is the lack of consideration for other risk factors for coronary artery disease such as follow-up lipid levels and controlled proportions of diabetes mellitus, as well as hypertension. These issues may impact TLR in patients with ILs. However, our study objective was to determine practical angiographic predictors in the study patients. The last limitation is regarding the relatively small sample size. In particular, repeated coronary angiograms are currently not performed very often due to very low in-stent restenosis via the use of drug-eluting stents, and thus, repeated angiograms are only performed in patients with recurrent anginas or ACS symptoms. Therefore, further studies in a wider study population are needed.

In conclusion, angiographic haziness and DS > 60% are independent angiographic predictors for TLR in patients with ILs. Our study is the first to show the angiographic findings of vulnerable plaques of ILs. However, further studies, such as IVUS or FFR studies, need to be strongly considered before decisions about the treatment of ILs are made when such angiographic features are observed.

## Figures and Tables

**Figure 1 biomedicines-12-02825-f001:**
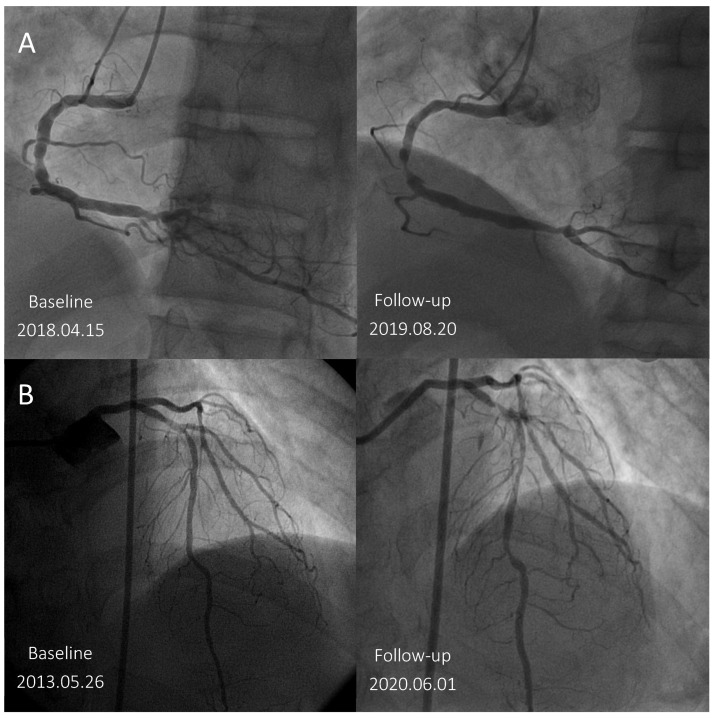
(**A**) ILs with angiographic haziness at the middle RCA progressed to severe stenosis with unstable angina and underwent revascularization at 16 months during the FU visit. (**B**) ILs with angiographic haziness at the proximal LAD progressed to severe stenosis with STEMI and needed revascularization at 84 months during the FU visit. FU, follow-up; IL, intermediate lesion; RCA, right coronary artery; LAD, left anterior descending artery; STEMI, ST elevation myocardial infarction.

**Figure 2 biomedicines-12-02825-f002:**
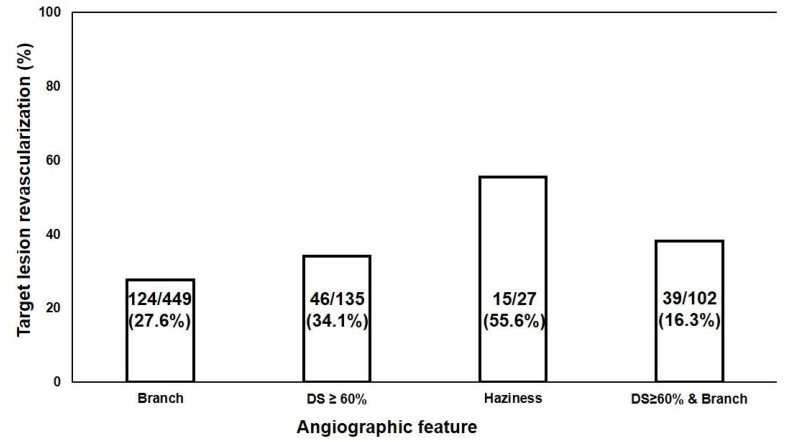
Proportion of angiographic features in patient with target lesion revascularization. DS, diameter stenosis.

**Figure 3 biomedicines-12-02825-f003:**
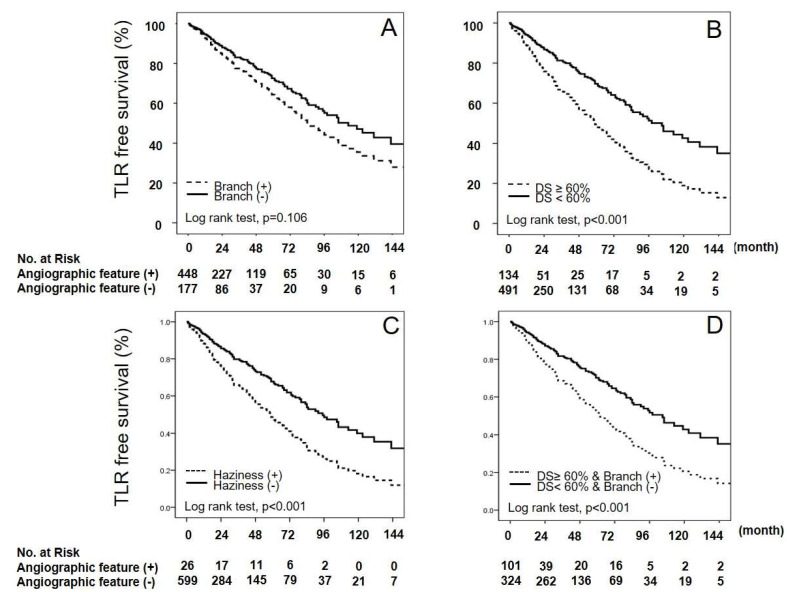
Kaplan–Meier curves for the presence of intermediate lesion features on the angiograms of a study group in terms of TLR-free survival during the follow-up period. TLR, target lesion revascularization; DS, diameter stenosis; (**A**–**D**) represent that TLR free survival curve according to the presence of each angiographic characteristic.

**Table 3 biomedicines-12-02825-t003:** Multivariable Cox proportional hazard analysis for TLR.

Total						
	Univariate Analysis			Multivariate Analysis		
Variables	HR	95% CI	*p*-Value	HR	95% CI	*p*-Value
% DS ≥ 60%	1.023	1.011–1.035	<0.001	1.025	1.013–1.037	<0.001
Haziness	1.855	1.087–3.165	0.023	2.126	1.240–3.644	0.006
Initial ACS	1.327	0.967–1.822	0.080			

TLR, target lesion revascularization; HR, hazard ratio; CI, confidence interval; % DS, percent diameter stenosis; ACS, acute coronary syndrome. This multivariable analysis included % DS ≥ 60%, haziness, initial ACS, and branches as the variables.

## Data Availability

The data that support the findings of this study are available from the corresponding author upon request. The data are not publicly available due to privacy or ethical restrictions.

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
