# Peer review of "Angiographic Predictors for Repeated Revascularization in Patients with Intermediate Coronary Lesions"

_biomedicines, 2024, doi:10.3390/biomedicines12122825_

Round 1

Reviewer 1 Report

Comments and Suggestions for Authors

Dear editor

I reviewed the article entitled “Angiographic Predictors for Revascularization in Patients with 2 Intermediate Coronary Lesions”. I congrats the authors for their interesting and original study. However, some revisions are required for the paper. My comments:

1- Abstract, methods, “non-randomized” word is not required. Please delete. Also, inclusion criteria should be defined more clearly in here.

2- were they stable coronary disease or acute coronary syndrome? The clinic of the patients should be defined. Also, detailed exclusion criteria should be presented.

3- what was the interobserver and intraobserver variability for degree of IL.

4- statistical analysis, the following sentence should be deleted. “Patient demographics and angiographic lesion 86 characteristics were compared according to the TLR.”

5- When the tables investigated, it can be seen that some of the continuous variables have not a normal distribution. Did you perform the normality test? You certainly perform a normality test, and if the variable that have not a normal distribution should be presented as median (Q1-Q3) and be compared with Mann whitney u test. Please help from a statistician

6-In the statistical analysis the authors stated that “Multiple stepwise linear regression analyses were performed to determine the clinical and angiographic correlates of outcome.” However, in the article I did not see the linear regression analysis!

7- Table 3, which variables entered into the multivariable model. Please detailed explain.

8- The authors stated in methods that patients between January 2008 56 and December 2020 were included. However, in Figure 1B, it is seen that there is a patients included in 2003.05.06. Please check

9- Discussion section is so limited. It should be expanded

Author Response

1- Abstract, methods, “non-randomized” word is not required. Please delete. Also, inclusion criteria should be defined more clearly in here.

  • We deleted “non-randomized” in the abstract and method section as well.
  • And changed from total 667 patients to total 667 consecutive patients

2- were they stable coronary disease or acute coronary syndrome? The clinic of the patients should be defined. Also, detailed exclusion criteria should be presented.

  • Diagnosis of the enrolled patients was described in the Table 1 which was expressed in red color.
  • Detailed exclusion criteria were described in the method section in red color which included patients without IL or planned PCI cases.

3- what was the interobserver and intraobserver variability for degree of IL.

  • Your comment about the inter/intraobserver variability for degree of IL is very important. We also thought it is very important. However, there was almost no disagreement between 2 researchers (YKK and SHK) regarding the IL. Furthermore, the senior researcher (JHB) gave second opinion if the 2 researchers showed different opinion about the degree of IL stenosis or IL characteristics. And also, there were no studies dealing with IL showed the variability regarding IL itself.
  • We changed/added the sentence in the method section in red color as follow to increase the reliability in estimating IL. Two researchers (YKK and SHK), who had more than 10 years` experience in coronary angiogram, reviewed and determined the angiographic stenosis with lesion characteristics. The senior cardiologist (JHB), who had more than 27 years` experience in coronary angiogram, gave second opinion if the 2 researchers showed different opinion about the degree of IL stenosis or IL characteristics.

4- statistical analysis, the following sentence should be deleted. “Patient demographics and angiographic lesion 86 characteristics were compared according to the TLR.”

  • We appreciate your comment very much. We deleted the sentence.

5- When the tables investigated, it can be seen that some of the continuous variables have not a normal distribution. Did you perform the normality test? You certainly perform a normality test, and if the variable that have not a normal distribution should be presented as median (Q1-Q3) and be compared with Mann whitney u test. Please help from a statistician

  • First of all, we appreciate your comment about it. We had discussed it with statistician. We draw a conclusion that the statistical method would be more adequate with independent t-test with these continuous variables because the study patients were 437 patients with 627 ILs and there is no reason to think that is not a normal distribution.

6-In the statistical analysis the authors stated that “Multiple stepwise linear regression analyses were performed to determine the clinical and angiographic correlates of outcome.” However, in the article I did not see the linear regression analysis!

  • The linear regression analysis result was used in the Table 3. And I described it more detailed which was also commented in your number 7 comment.

7- Table 3, which variables entered into the multivariable model. Please detailed explain.

  • We added the following sentence in the Table 3. This multiple analysis included %DS≥60%, haziness, initial ACS and branch as variables.

8- The authors stated in methods that patients between January 2008 56 and December 2020 were included. However, in Figure 1B, it is seen that there is a patients included in 2003.05.06. Please check

  • There was an error in making figure. We corrected the date in figure 1B.

9- Discussion section is so limited. It should be expanded

  • We appreciate your comment for our manuscript. We expand the discussion section and described in red color in the discussion section.
  • Currently, FFR is a very important tool in decision making about IL regarding the need for revascularization. [3] But most IL have FFR value more than 0.8 and also this FFR negative IL cannot guarantee the safety of patients. [7] Because most ACS can be resulted from plaque rupture in non-stenotic coronary lesion. Thus, FFR itself has some limitation to see the future risk of IL, although it is very helpful in deciding the current ischemia of IL.

Reviewer 2 Report

Comments and Suggestions for Authors

I read with interest the study by Yong Kyun Kim et al. The authors attempted to pinpoint angiographic factors potentially associated with higher risk of repeat revascularization.

I have a few major comments.

The quality of English throughout the manuscript is poor and must be substantially improved. The text must be proof-red. On occasions the point the authors wish to make is not clear or whole phrases make no sense.

The title is not complete. Essentially the authors looked for angiographic predictors for repeat revascularization. A title like: 'Angiographic Predictors for Repeat Revascularization in Patients with Intermediate Coronary Lesions' would be more appropriate.

The sample size is relatively small in this retrospective study spanning 12 years hence, extreme caution should be exercised in interpreting the results.

I have concerns about the angiographic variables, in particular the ulcer/erosion, calcification and haziness. In the era of intracoronary imaging, the previously mentioned angiographic findings are notoriously non-sensitive and non-specific with significant interobserver variability.

I would stop at this juncture and summarize saying that although the original concept of the study is almost brilliant, the execution is subpar. However, I would be happy to support their work if they could address the above-mentioned remarks.

Comments on the Quality of English Language

Poor quality of English, the whole manuscript must be proof-red.

Author Response

The quality of English throughout the manuscript is poor and must be substantially improved. The text must be proof-red. On occasions the point the authors wish to make is not clear or whole phrases make no sense.

  • We are very sorry for the problems. We will do professional language editing service.

The title is not complete. Essentially the authors looked for angiographic predictors for repeat revascularization. A title like: 'Angiographic Predictors for Repeat Revascularization in Patients with Intermediate Coronary Lesions' would be more appropriate.

  • We appreciate your valuable comment for our paper. We revised the title as you commented.

The sample size is relatively small in this retrospective study spanning 12 years hence, extreme caution should be exercised in interpreting the results.

  • We agree your comment about it. The sample size consisted of the patients who underwent at least 2 coronary angiograms during the 12 years. The repeat coronary angiogram was done only in patients who had recurrent angina or ACS symptoms. TLR rate was very low in the current DES era. Therefore, the study population was relatively small in our study. Nevertheless, we could get the significant angiographic predictors of IL for repeat revascularization. But, there are several limitations especially with the sample size. So, we added the following sentence in the limitation section.
  • The last limitation is regarding the relatively small sample size. However, instent restenosis is quite low in the current drug eluting stent era and the repeat angiogram is currently done only in patients with recurrent angina or ACS symptoms.

I have concerns about the angiographic variables, in particular the ulcer/erosion, calcification and haziness. In the era of intracoronary imaging, the previously mentioned angiographic findings are notoriously non-sensitive and non-specific with significant interobserver variability.

  • I totally agree with your comment. IVUS examination is a crucial in determining those characteristics. But, IVUS penetration rate is still quite low in the real world although IVUS predictors for repeat revascularization was reported in several studies and the purpose of our study is to find the angiographic predictors for repeat revascularization in IL. In order to decrease the variability in assessment of IL, we did the following effort and described it in the method section.
  • Two researchers (YKK and SHK), who had more than 10 years` experience in coronary angiogram, reviewed and determined the angiographic stenosis with lesion characteristics. The senior cardiologist (JHB), who had more than 27 years` experience in coronary angiogram, gave second opinion if the 2 researchers showed different opinion about the degree of IL stenosis or IL characteristics.

I would stop at this juncture and summarize saying that although the original concept of the study is almost brilliant, the execution is subpar. However, I would be happy to support their work if they could address the above-mentioned remarks.

  • We appreciate your very important comments about our paper. We tried/revised it fully according to your comments. We will happy if you consider again our paper for potential publication in Biomedicines.

Round 2

Reviewer 1 Report

Comments and Suggestions for Authors

Dear author, thank for your responses. However, Table 3 is not linear regression analysis. It is logistic regression analysis. this should be correected.

Author Response

Dear author, thank for your responses. However, Table 3 is not linear regression analysis. It is logistic regression analysis. this should be corrected.

  • We corrected it from ‘multiple linear regression’ to ‘logistic regression’ in the method section.
  • Thank you very much for your kind comments and help for our article for the improvement.

Reviewer 2 Report

Comments and Suggestions for Authors

I commend the authors for reverting with the new version of the manuscript. This is an improved version of the original manuscript which communicates more clear messages supported by the provided data.

I would support the publication of this paper.

Author Response

I commend the authors for reverting with the new version of the manuscript. This is an improved version of the original manuscript which communicates more clear messages supported by the provided data.

I would support the publication of this paper.

  • We appreciate your second review for our article very much.
  • And thank you very much for your kind understanding and it will definitely encourage our enthusiasm to the clinical study.